# Trichosanthin Promotes Anti-Tumor Immunity through Mediating Chemokines and Granzyme B Secretion in Hepatocellular Carcinoma

**DOI:** 10.3390/ijms24021416

**Published:** 2023-01-11

**Authors:** Kaifang Wang, Xiaona Wang, Minghuan Zhang, Zhenguang Ying, Zeyao Zhu, Kin Yip Tam, Chunman Li, Guowei Zhou, Feng Gao, Meiqi Zeng, Stephen Cho Wing Sze, Xia Wang, Ou Sha

**Affiliations:** 1School of Dentistry, Shenzhen University Medical School, Shenzhen 518000, China; 2Department of Biology, Faculty of Science, Hong Kong Baptist University, Hongkong 999077, China; 3Department of Anatomy and Histology, School of Basic Medical Sciences, Shenzhen University Medical School, Shenzhen 518000, China; 4Department of Biology, School of Life Sciences, Southern University of Science and Technology, Shenzhen 518000, China; 5Faculty of Health Sciences, University of Macau, Macau, China; 6Guangdong Provincial Key Laboratory of Infectious Diseases and Molecular Immunopathology, Shantou University Medical College, Shantou 515000, China

**Keywords:** Trichosanthin (TCS), hepatocellular carcinoma (HCC), T cell, chemokine, Granzyme B (GrzB), apoptosis

## Abstract

Trichosanthin (TCS) is a type I ribosome-inactivating protein extracted from the tuberous root of the plant *Trichosanthes*. TCS shows promising potential in clinical drug abortion, anti-tumor and immunological regulation. However, the molecular mechanisms of its anti-tumor and immune regulation properties are still not well discovered. In the present study, we investigated the anti-tumor activity of TCS in hepatocellular carcinoma (HCC), both in vitro and in vivo. Both HCC cell lines and xenograft tumor tissues showed considerable growth inhibition after they were treated with TCS. TCS provoked caspase-mediated apoptosis in HCC cells and xenograft tumor tissues. The recruitment of CD8^+^ T cells to HCC tissues and the expression of chemokines, CCL2 and CCL22, were promoted upon TCS treatment. In addition, TCS induced an upregulation of Granzyme B (GrzB), TNF-α and IFN-γ in HCC tissues, which are the major cytotoxic mediators produced by T cells. Furthermore, TCS also resulted in an increase of mannose-6-phosphate receptor (M6PR), the major receptor of GrzB, in HCC tissues. In summary, these results suggest that TCS perhaps increases T-cell immunity via promoting the secretion of chemokines and accelerating the entry of GrzB to HCC cells, which highlights the potential role of TCS in anti-tumor immunotherapy.

## 1. Introduction

Hepatocellular carcinoma (HCC) is the second most common cause of cancer mortality worldwide [1]. Since HCC is highly aggressive and metastatic, only about 10% of patients have limited options, such as surgical resection, liver transplantation and local ablation [2]. HCC is also one of the most common chemotherapy-resistant tumors. The continuous administration of conventional chemotherapeutic agents and antitumor immune agents causes side effects, such as tumor resistance and poor prognosis. Therefore, it is imperative to explore new drugs or therapeutic strategies targeting HCC [3]. Researchers have identified numerous plant-derived extracts with potent antitumor properties, and representative ones include paclitaxel [4], curcumin [5], millipedium [6], Trichosanthin [7], etc.

Trichosanthin (TCS), a single-chain ribosome-inactivating protein extracted from the tuberous root of the traditional Chinese herb *Trichosanthes*, exhibits prospective application in clinical drug abortion, anti-virus, anti-tumor and immune regulation [8]. Numerous studies have shown that TCS could directly inhibit the proliferation and apoptosis of cancer cells by regulating the expression of Bcl-2 [9], inducing S-phase cell cycle arrest in cancer cells [10], inhibiting tumor dysplasia-related signaling pathways [11], increasing the expression or activation of caspase family proteins [12,13,14], etc. In physiological conditions, TCS could regulate the immune status of the body by regulating the CD4^+^/CD8^+^ T-cell ratio and producing related immune cytokines in peripheral blood [15]. Studies have shown that TCS could enhance the sensitivity of tumors to chemotherapeutic drug treatment [13,16]. Recombinant TCS has shown potent anti-tumor effects [17,18,19]. It is crucial to investigate the anti-tumor mechanisms and potential applications of TCS.

Granzyme B (GrzB) is an extremely high anti-tumor bioactive protein produced mainly by CD8^+^ T cells and NK cells [20]. Numerous studies have shown that GrzB can rapidly activate caspase 3-related signaling pathways in target cells [21], which in turn promote cancer cell apoptosis or inflammatory death [22]. Our previous works on immunodeficient nude mice found that the combination of TCS and GrzB had a positive effect in inhibiting HCC, and TCS enhanced the translocation of GrzB from mannose-6-phosphate receptors (M6PR) to HCC cells [23]. However, whether TCS could inhibit HCC by regulating anti-tumor immunity has not been examined yet. In this study, we used TCS to treat HCC cells and a xenograpft tumor model to investigate the mechanism of TCS regulating the recruitment of T cells in the host immune response against HCC.

## 2. Results

### 2.1. TCS Reduces the Viability of HCC Cells in Culture

To test if TCS was able to inhibit HCC cell growth in culture, TCS (concentration ranged from 1.5625 to 400 μg/mL) was administered to the H22 HCC cell line for 24, 48 and 72 h. TCS inhibited H22 cell viability in a dose-dependent manner (Figure 1A). The IC50 of HCC cells treated with TCS for 48 h and 72 h was approximately 25 μg/mL (Figure 1A). Then, we treated HCC cells with 25 μg/mL TCS and assayed cell viability at multiple time points, and we observed a significant decrease of cell viability after 36 h (Figure 1B). The Calcein-AM/PI assay also showed a significant and time-dependent increase of dead HCC cells after TCS treatment (Figure 1C). PARP, a nucleus polymerase that appears to be involved in DNA repair and that is a common apoptosis marker cleaved by Caspase-3 [24], was also induced by TCS in a dose-dependent manner (Figure 1F). To further confirm TCS might impair cell viability, the apoptosis inhibitor Z-VAD-FMK was applied to treat HCC cells for 48 h, combined with TCS at multiple concentrations. Z-VAD-FMK significantly inhibited the cell death and PARP cleavage induced by TCS (Figure 1E,F). This suggested that TCS triggered HCC cell death mainly by promoting caspase activities.

### 2.2. TCS Promoted HCC Cell Death via Apoptosis

Apoptosis and autophagy are common modes of tumor cell death [25,26]. It has been reported that TCS can promote the death of Oral squamous cell carcinoma SCC25 by inducing cell apoptosis [9]. In order to better understand the mechanisms of TCS-induced cell death, HCC cells were treated with 25 μg/mL TCS, total protein was extracted and the expression levels of apoptosis- and autophagy-related proteins were detected by Western blot. Apoptosis usually involves the activation of a series of caspase enzymes. The upstream caspase of the intrinsic pathway is caspase 9, while the exogenous pathway is caspase 8; after that, internal and external pathways converge to caspase 3 [27]. Western blot analysis showed that Caspase 9, Caspase 8 and Caspase 3 were all decreased in HCC cells after 72 h of TCS treatment (Figure 2A). Instead, the level of Cleaved-caspase 9, Cleaved-caspase 8 and Cleaved-caspase 3 were all elevated after 48 h of TCS treatment (Figure 2A).

Studies have also shown that TCS can inhibit the growth of gastric cancer cell MKN-45 by inducing autophagy [28]. In the process of autophagosome formation, LC3I is lipidized to form LC3II; therefore, LC3I/LC3-II is considered a marker of autophagosome. In addition, the autophagic receptor p62 is also commonly used as an autophagic marker [29]. However, our results show that the level of the autophagy markers P62 and LC3I/LC3II were not significantly different between the control and TCS-treated groups (Figure 2B). This indicated that TCS did not induce significant autophagy in HCC cells. Therefore, TCS induced HCC cell death, mainly via apoptosis.

### 2.3. TCS Inhibits HCC Tumor Growth In Vivo

To investigate the therapeutic effect of TCS on HCC in vivo, an H22 HCC xenograpft model was established subcutaneously in BALB/c mice, which were treated with TCS at Day 5, 7, 9, 11, 13, 15 and 17 (Figure 3A). TCS treatment significantly inhibited the growth of HCC tumors in mice in a dose-dependent manner (Figure 3B). The application of TCS at the highest concentration of 2 μg/g achieved a tumor volume inhibition rate of about 52.91% (Figure 3C) and a tumor mass inhibition rate of about 55.01% (Figure 3D,E). Although the mice treated with TCS exhibited a decreasing trend in body weight, there was no significant difference compared to the control group. These results suggested that TCS could inhibit HCC cell growth in vivo.

Next, we assayed Ki67, a tumor proliferation marker [30], by immunohistochemical fluorescence. The number of Ki67-positive cells decreased in the TCS-treated group in a dose-dependent manner (Figure 4A). TCS significantly promoted the activation of Caspase 9, Caspase 8 and Caspase 3 in tumor tissues. The levels of Cleaved-caspase 9, Cleaved-caspase 8 and Cleaved-caspase 3 were significantly increased (Figure 4B). Therefore, TCS could inhibit the progression of HCC malignancy in mice, mainly through activation of the caspase pathway. 

### 2.4. TCS Promotes Infiltration of CD8^+^ T Cells into HCC

TCS promoted CD8^+^ T-cell infiltration in cancer tissues [15]. Therefore, we investigated the level of the infiltration of CD8^+^ T cells in mouse HCC xenograft tissues. The number of CD8^+^ T cells in HCC tumor tissues increased with the doses of TCS (Figure 5A). Interestingly, enrichment of CD8^+^ T cells could be seen at the edge of tumor tissues (Figure 5B).

Chemokines are signaling molecules necessary for normal T-cell transport and function [31], and the interleukin family plays an important role in immune regulation and inflammatory responses [32]. In addition, the increased secretion of TNF-α and IFN-γ contributes to the antitumor interaction with T cells [33]. Therefore, mRNA expressions of chemokines CCL2, CCL17 and CCL22, as well as IL-6, IL-18, TNF-α and IFN-γ, in tumor tissues and H22 cells were detected by RT-qPCR. The results of RT-qPCR showed that the expression levels of CCL2, CCL22, TNF-α and IFN-γ in tumor tissues were significantly increased after TCS treatment. Additionally, CCL2, CCL17, CCL22 and TNF-α were also increased in H22 cells after TCS treatment (Appendix A). Furthermore, according to the results of RT-qPCR, four cytokines with significant differences, CCL2, CCL22, TNF-α and IFN-γ, were detected for protein levels by ELISA. The results of tumor tissue samples showed that the protein levels of chemokines CCL2 and CCL22, as well as TNF-α and IFN-γ, were significantly increased upon TCS treatment (Figure 5C). Serum levels of CCL22, TNF-α and IFN-γ were elevated after TCS treatment, although only IFN-γ statistically significantly increased (Figure 5D). In addition, the expression levels of chemokines CCL2 and CCL22 in HCC cell culture fluid were also significantly increased upon TCS treatment (Figure 5E).

These results suggested that TCS could enrich CD8^+^ T cells to tumor tissues and promote the expression of chemokines in HCC cells, which would enhance the anti-tumor immune response of the organism.

### 2.5. TCS Enhances the Expression of Granzyme B and M6PR

GrzB, a serine proteinase released by cytotoxic T cells and NK cells, mediates cell apoptosis in target cells [34,35]. Since TCS could recruit CD8^+^ T cells to xenograft tumor tissues, we next examined whether the expression and transportation of GrzB were affected by TCS. As expected, GrzB was elevated in HCC tissues in TCS-treated mice in a dose-dependent manner (Figure 6A–C). The proportion of TUNEL^+^/GrzB^+^ cells in tumor tissues significantly increased as TCS dosages were raised (Figure 7A–C). We also confirmed that the alterations in the number of TUNEL^+^ and TUNEL^+^/GrzB^+^ cells were positively correlated with TCS dosages (Figure 7B,C). Our previous studies showed the translocation of GrzB from mannose-6-phosphate receptors (M6PR) to HCC cells was enhanced by TCS [23]. Therefore, we next examined the level of M6PR, both in cell lines and xenograft tumor tissues. We showed that TCS promoted the expression level of M6PR in tumor tissues and HCC cells (Figure 8C,D). As TCS dosages were augmented, the proportion of TUNEL^+^/M6PR^+^ cells in tumor tissues also significantly increased (Figure 8A,B). The number of M6PR and TUNEL double-positive cells showed a positive correlation with the TCS dose (Figure 8A,B). These data indicated that TCS could inhibit HCC growth by inducing the upregulation of GrzB and promote HCC cell apoptosis in vivo by encouraging M6PR to deliver GrzB into HCC cells.

## 3. Discussion

TCS is the major active ingredient of *Trichosanthes Kirilowii* [36]. Previous studies reported TCS alone showed an excellent inhibitory effect on cancer cell proliferation in vitro [9,23,37]. However, TCS alone had not significantly inhibited tumor growth in immunodeficient nude mice in vivo [23]. In the present study, we constructed tumor models in BALB/c mice with a functional immune system. We found that TCS not only activated caspase family proteins in tumor tissues, but also promoted T-cell immunity. Chemotactic enrichment of T cells in HCC tissues and elevated levels of GrzB were observed in vivo. This suggests that TCS has considerable promise as an immunotherapy tool and may increase the effectiveness of anti-tumor treatments.

TCS induced cell cycle block, autophagic death and caspase-mediated apoptosis in cancer cells [28,38]. Proper levels of autophagy remove damaged organelles from cells and contribute to the maintenance of normal cell survival, while inducing excessive autophagy causes cell death [39,40]. TCS induced ROS production in gastric cancer cells, which in turn promoted the autophagic death of gastric cancer cells [28]. However, in the present study, TCS barely promoted the autophagic death of HCC cells. Recently, Hu et al. detected by proteomics that TCS inhibited nuclear proliferation factors in human choriocarcinoma cell lines, mainly inducing caspase-mediated apoptosis in cancer cells [41]. Caspase 8 is a key protease in apoptosis caused by exogenous factors [42]. Caspase 9 is an endogenous apoptosis-related protease activated by damage to organelles, such as endoplasmic reticulum or mitochondria [43]. Both cleaved-caspase 8 and cleaved-caspase 9 activate caspase 3 to form cleaved-caspase 3, which in turn degrades the DNA repair-associated protein PARP and induces apoptosis in cancer cells [44,45]. The results of our study showed that apoptosis inhibitor (Z-VAD-FMK) was able to inhibit TCS-induced HCC cell death, and caspase proteins were significantly activated. Thus, TCS mainly activated caspase-mediated endogenous and exogenous apoptotic pathways to inhibit HCC cells.

TCS can regulate the immune functions of macrophages [46], DC cells [47] and T cells [48]. However, variable effects of TCS on the regulation of immune cell function have been observed in different diseases, such as physiological conditions [48], inflammation [46], HIV infection [49] and cancer [15]. Enhancement of the antitumor response of T cells would be one of the effective ways to inhibit HCC [50]. T cells can induce cancer cell death through both receptor and non-receptor mediated pathways [51]. TCS could enhance the immune effect of T cells against lung cancer through the receptor pathway by enhancing the expression of class I, restricted, T cell-associated molecules in CD8^+^ T cells [15]. A recent study showed that tonics containing the Chinese herb *Trichosanthes* significantly elevated the serum levels of cytokines, such as IFN-γ, IL-6 and TNF-α, which would be beneficial in enhancing the immune effect of the lymphocyte system [52]. Numerous immune subpopulations, including T cells, natural killer (NK) cells and B cells, can produce IFN-γ in the tumor microenvironment [53]. CD8^+^ cytotoxic T lymphocytes (CTL) are known to be the main producers of IFN-γ and one of the indicators of activation of the T-cell antitumor function [54]. Studies have shown that IFN-γ could induce apoptosis or scorch death of cancer cells through IFN-γ receptors on the surface of cancer cells [55,56]. TNF-α plays different roles in the pre-cancerous and cancer microenvironments. Despite a sustained inflammatory response possibly being detrimental to suppressing precancerous lesions, increased TNF-α in a tumor microenvironment could effectively activate TNFR1 to trigger cancer cell suppression [57,58]. Tumor-infiltrating lymphocytes are an important prognostic factor for cancer progression and a key player in cancer immunotherapy. CCL2, CCL17 and CCL22 are cytokines with a role in T-cell recruitment [59,60]. Despite the role of CCL2 in recruiting both cytotoxic T cells (CTL) and monocytes to tumor sites [61,62], studies have shown that enhancement of the CCL2/CCR2 axis [63] or inhibition of CCL2 nitration [64] in antitumor therapy significantly promotes T-cell infiltration and exerts antitumor effects. Binding of CCL22 with CCR4 enhances T-cell dendritic cell binding and increases CTL activation [65], while enhancing tumor cell responses to IFN-γ [66]. GrzB, the cytokine secreted by CD8^+^ T cells, has the greatest killing effect on cancer cells by directly or indirectly activating caspases to inhibit cancer cell proliferation and induce apoptosis [67]. GrzB could be transported by M6PR into cancer cells [68]. Previous studies by our group demonstrated that the combination of TCS and GrzB in the treatment of nude mice implanted with HCC significantly inhibited the growth of HCC, and TCS promoted the transport of GrzB by M6PR into HCC cells [23]. In the present study, we examined the anti-tumor mechanisms of TCS by facilitating the xenograft tumor model within BALB/c mice with a functional immune system. We observed that CD8^+^ T cells were recruited into the HCC tissues in the TCS-treated group in a dose-dependent manner. The expression level of chemokines was elevated in HCC tissues, a favorable condition for T cells to kill HCC cells through the receptor or cytokine pathway. Additionally, TCS directly elevated GrzB levels in tumor tissues, which were significantly and positively correlated with apoptotic cancer cells. This implies that TCS can enhance T-cell anti-tumor immunity against HCC by encouraging T-cell enrichment, elevating the expression of chemokines in tumor cells and promoting their production of GrzB. These results provide supporting information for the study of TCS to enhance anti-tumor immunity.

## 4. Materials and Methods

Recombinant TCS protein extraction and purification: The recombinant TCS plasmid was constructed by Tsingke Biotechnology Co., Ltd. (Guangzhou, China). The rTCS sequence was inserted between the EcoRI site and the XhoI site of pet-28a+. *E.coli* BL21 (DE3) was used to induce the expression of recombinant TCS protein (when OD600 = 0.60, add a final concentration of 2 mM IPTG). Centrifuge the prokaryotic expression induced bacterial solution at 4 °C, 12,000 rpm, 2 min; discard the supernatant; then, add the lysis buffer (50 mM NaH2PO4, 300 mM NaCl, 10 mM imidazole, pH 8.0); blow repeatedly; and perform ultrasonic lysis on ice (power: 30%, ultrasound: 5 s, stop: 10 s, total working time: 30 min); and then collect the supernatant at 4 °C, 12,000 rpm, 30 min. After the supernatant containing the target protein is filtered and sterilized, it is purified by the protein purifier (GE, AKTA purifier, Boston, MA, USA). The obtained recombinant TCS protein was freeze-dried and stored at −30 ℃.

Cell culture: Mouse HCC cell line (H22) (Procell, CL-0341) was cultured in Roswell Park Memorial Institute 1640 (RPMI-1640) medium containing 10% fetal bovine serum, 100 U/mL of penicillin and 100 μg/mL of streptomycin. HCC cells were cultured in a 5% CO_2_ incubator at a constant temperature of 37 °C.

HCC cell viability assay: H22 cells were inoculated in 96-well plates at a density of 5 × 10^3^, with 100 μL cell culture solution per well. For a concentration-dependent assay, TCS (0, 1.5625, 3.125, 6.25, 12.5, 25, 50, 100, 200 and 400 μg/mL) was added to the treatment for 24 h, 48 h and 72 h. For a time-dependent assay, TCS (0, 25 μg/mL) was added to the treatment for 0 h, 12 h, 24 h, 36 h, 48 h, 60 h and 72 h. Cell viability was then determined by the method of CCK-8. Briefly, at the end of the intervention, 10 µL of CCK-8 solution was added to each well of the 96-well plate and mixed, and the reaction plates were incubated at 37 °C for 1–3 h. The absorbance of each well solution was measured, using the microplate reader (BioTek, Epoch, Winooski, VT, USA) at 450 nm wavelength detection light.

Calcein-AM/PI staining: Unspecific esterases present in living cells can metabolize Calcein-AM and emit green fluorescence. Propidium iodide (PI) will label the nuclei of dead cells. H22 cells were inoculated in 6-well plates at a concentration of 1 × 10^5^/mL culture medium per well. Cells were treated with different doses of TCS, then Calcein-AM and PI were added and incubated for 5 min, and fluorescent pictures were collected under a confocal microscope. Cell death index calculation: percentage of dead cells = number of dead cells/total number of cells.

Antibodies: The antibody information in this study is shown in Appendix A.

Reagents: Z-VAD-FMK (Beyotime, Nanjing, China, cat# C1202), Calcein-AM (MCE, Monmouth Junction, NJ, USA, cat# HY-D0041), PI (MCE, Monmouth Junction, NJ, USA, cat# HY-D0815), DAPI (Solarbio, Beijing, China, cat# D8200) DeadEnd™ Fluorometric TUNEL System (Beyotime, Nanjing, China, cat# C1088). IHC Reagent Kit (Solarbio, Beijing, China, cat# SP0021), Total Mrna Extraction Kit (Promega, Madison, WI, USA, cat# LS1040), Hifair III 1st Strand cDNA Synthesis SuperMix for qPCR (gDNA digester plus) (Yeasen, Shanghai, China, cat# 11141ES60), GoldenstarTM RT6 cDNA Synthesis Kit (TSINGKE, Xi’an, China, cat# TSG302M), ELISA MAX™ Standard Set Mouse MCP-1 (BioLegend, San Diego, CA, USA, cat# 432701), ELISA MAX™Deluxe Set Mouse IFNγ (BioLegend, San Diego, CA, USA, cat# 430804), Mouse TNF-α ELISA kit (Jianglaibio, Shanghai, China, cat# JL10484), Mouse MDC/CCL22 ELISA kit (Jianglaibio, Shanghai, China, cat# JL11125), etc.

Apoptosis inhibition test assay: Z-VAD-FMK is a pan-caspase inhibitor that prevents the cleavage degradation of DNA repair enzyme PARP by Caspase family proteins [69]. H22 cells were inoculated in 6-well plates at a concentration of 1 × 10^5^/mL culture medium per well. Z-VAD-FMK was intervened according to the following protocol: control (no TCS), TCS (12.5 μg/mL), TCS (25 μg/mL), TCS (50 μg/mL), TCS (12.5 μg/mL) + Z-VAD-FMK (40 μM), TCS (25 μg/mL) + Z-VAD-FMK (40 μM), TCS (50 μg/mL) + Z-VAD-FMK (40 μM) for 48 h of continuous intervention, and cells were analyzed by Calcein-AM/PI staining and immune-blotting for PARP-related proteins.

Xenograft tumor model: The procedure of the animal experiments performed in the study fulfilled the requirements of the ethical review committee. Male BALB/c mice (5-week-old, 20 ± 2 g, Guangdong Medical Laboratory Animal Center, Guangzhou) were housed in an SPF-grade environment with a 12 h light/dark cycle and maintained on free diets. Mice were injected subcutaneously with 5 × 10^5^ H22 HCC cells in the right axilla. Beginning on the fifth day after implantation, mice with H22 HCC were randomly divided into four groups of five mice each and treated according to the following treatment protocols: phosphate-buffered saline (PBS) control group, 0.5 μg/g (TCS weight/mouse body weight) TCS group, 1 μg/g body weight TCS group, 2 μg/g body weight TCS group. Body weight TCS group. Control group was injected with 100 μL of PBS, and drug treatment was injected on the 4, 6, 8, 10, 12, 14 and 16 days after implantation of HCC cells. Tumor volume and body weight were measured, and tumor size was estimated according to the following formula: tumor volume = 0.5 × maximum diameter × shortest diameter × shortest diameter.

Immunohistochemistry (IHC) assay: Immunohistochemical staining of mouse tumor tissues was performed using the IHC kit and GrzB polyclonal antibody. Tumor tissues embedded in paraffin blocks were cut into 5 μm thick sections. The paraffin sections were dewaxed and hydrated, following 3% hydrogen peroxide treatment and antigen repair boiling in sodium citrate solution for 10 min. After incubation with normal goat serum at room temperature for 20 min, the tissue sections were incubated with anti-GrzB antibody (1:500) overnight at 4 °C. Then, samples were incubated with biotinylated goat anti-rabbit serum immunoglobulin G (IgG) antibody (1:100) for 30 min at 37 °C. After thorough washing, streptavidin-POD working solution was added and incubated for 30 min at 37 °C. Sections were counter-stained with hematoxylin. Morphological images were collected using an Olympus microscope. For immunofluorescence assay, tissue sections were incubated with antibodies against Ki67 (1:500), M6PR (1:500), GrzB (1:500) and CD8 (1:500) overnight at 4 °C. Samples were then incubated with appropriate FITC, Alexa Fluor 555 or Horseradish Peroxidase-conjugated secondary antibodies for 3 h. Nuclei were stained with DAPI or hematoxylin, followed by observation and image capture under a confocal microscope.

TUNEL apoptosis assay: Tumor tissues were examined using the Beyotime one-step TUNEL apoptosis assay kit. After dewaxing and hydration of the tumor tissues, the TUNEL staining procedure was performed, according to the manufacturer’s instructions. The nuclei were stained with DAPI.

Western blot assay: Protein extraction and Western blot assay were performed, as described in the previous report [23]. The following antibodies were used for detection: anti-Caspase 3 (1:1000), anti-Caspase 8 (1:1000), anti-Caspase 9 (1:1000), anti-Cleaved-caspase 3 (1:1000), anti-Cleaved-caspase 8 (1:1000), anti-Cleaved-caspase 9 (1:1000), anti-GrzB (1:1000), anti-M6PR (1:1000), anti-LC3B (1:1000), anti-P62 (1:1000), anti-GAPDH (1:5000). They were incubated overnight at 4 °C and then incubated with horseradish peroxidase (HRP)-conjugated counterpart secondary antibody (1:5000) at room temperature. Chemiluminescence was performed using ECL Ultra HRP substrate and photographed under the SAGECREATION ChemiMini™ Imaging System.

Quantitative real-time PCR: Total RNA from HCC cells and tumor tissues was extracted using the Total RNA Extraction Kit. Using the GoldenstarTM RT6 cDNA Synthesis Kit, mRNA was reversed into cDNA by reverse transcription procedure. Quantitative PCR amplification was performed using the Hifair III 1st Strand cDNA Synthesis Super Mix for qPCR in Quantstudio™ 7 Flex Real-Time PCR System (ABI). The expression of target genes was normalized against GAPDH using the 2^−ΔΔCt^ assay. Primer oligos were synthesized by TSINGKE Biological Co., Ltd. (Beijing, China) and are listed in Appendix A.

Enzyme linked immunosorbent assay (ELISA): Cell culture supernatants, mouse serum and tumor tissues were collected. Then, the levels of TNF-α, IFN-γ, CCL22 and CCL2 were assessed using ELISA kits, according to the manufacturer’s instructions. Absorbance was measured using a microplate reader (BioTek Instruments, Inc., Winooski, VT, USA).

Statistical analysis: Statistical analyses were performed using SPSS 26.0 software (SPSS Inc., Chicago, IL, USA). The error bars in the graphical data represent means ± Standard Error of Mean (SEM). Three or more comparisons were compared using one-way ANOVA, followed by the least significant difference (LSD) test. Unpaired two-tailed Student’s t-test was used to compare two sets of data. A value of *p* < 0.05 was considered a statistically significant difference between the data groups. *, *p* < 0.05; **, *p* < 0.01; ***, *p* < 0.001.

## 5. Conclusions

In conclusion, the results of this study demonstrated that TCS inhibited HCC cells by activating caspase, recruiting CD8^+^ T cells, enhancing the expression of chemokines and up-regulating M6PR genes to transport GrzB (Figure 9). This also indicates that TCS is a natural drug with great potential to enhance anti-tumor immunity, which is valuable for further in-depth pharmacological studies.

## Figures and Tables

**Figure 1 ijms-24-01416-f001:**
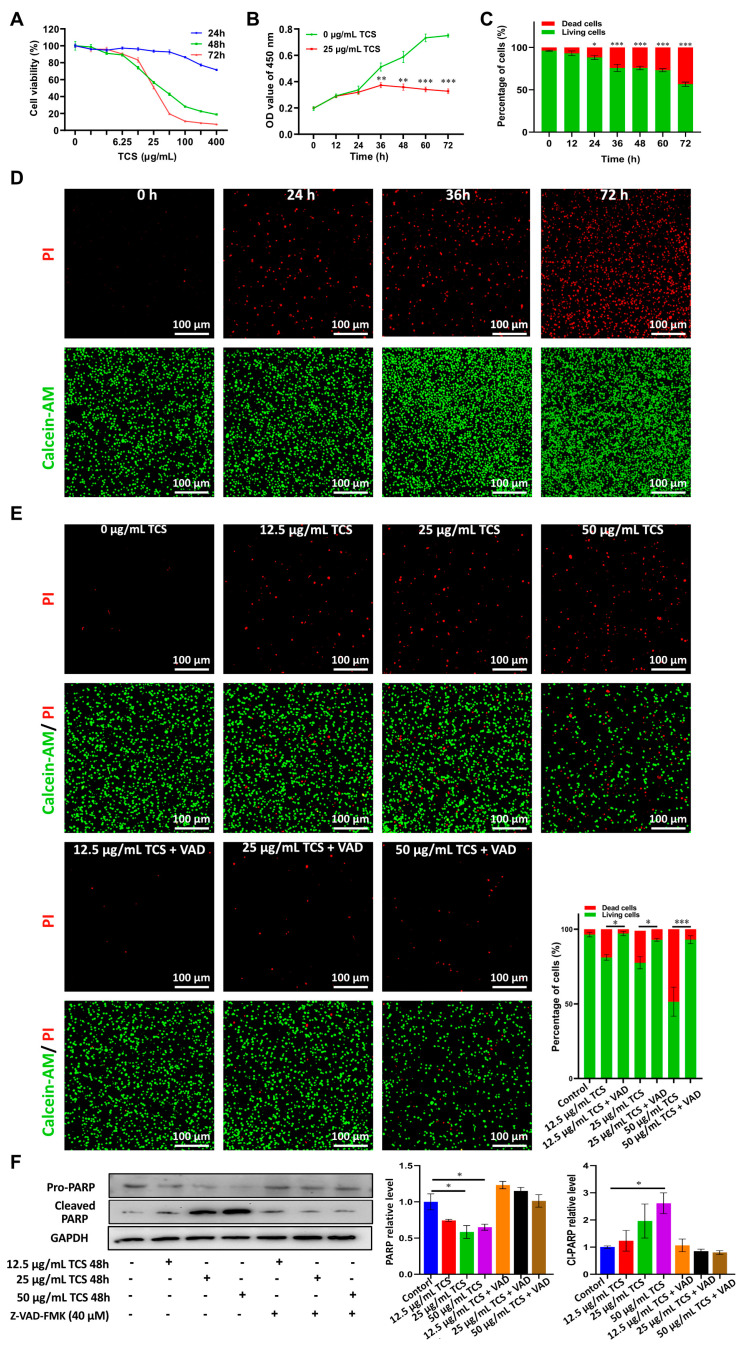
Effects of TCS on the cell viability and death of HCC cells. (**A**) H22 HCC cells were treated with different doses of TCS for 24 h, 48 h and 72 h. CCK-8 assay with absorbance at 450 nm was used to evaluate the cell viability; (**B**) IC50 dose of TCS (25 μg/mL) was used to treat H22 HCC cells for 12 h, 24 h, 36 h, 48 h, 60 h and 72 h. CCK-8 assay with absorbance at 450 nm was used to evaluate the cell viability; (**C**) 25 ug/mL TCS treated HCC cell lines at different times with the ratio of dead cells to live cells; (**D**) IC50 dose of TCS (25 μg/mL) was used to treat H22 HCC cells for 0 h, 24 h, 36 h and 72 h. The Calcein-AM/PI method was used to detect dead or alive cells, with green as live cells and red as dead cells. Bar = 100 μm; (**E**) H22 HCC cells were treated with different doses (0, 12.5, 25 and 50 μg/mL) of TCS. Meanwhile, 40 μM caspase inhibitor (Z-VAD-FMK) was used in combination. Calcein-AM/PI assay was used to detect dead or alive cells; green is live cells, red is dead cells. Bar = 100 μm; (**F**) Western blot assay for PARP and Cleaved-PARP protein expression after 48 h of TCS and Z-VAD-FMK coadministration. *, *p* < 0.05; **, *p* < 0.01; ***, *p* < 0.001.

**Figure 2 ijms-24-01416-f002:**
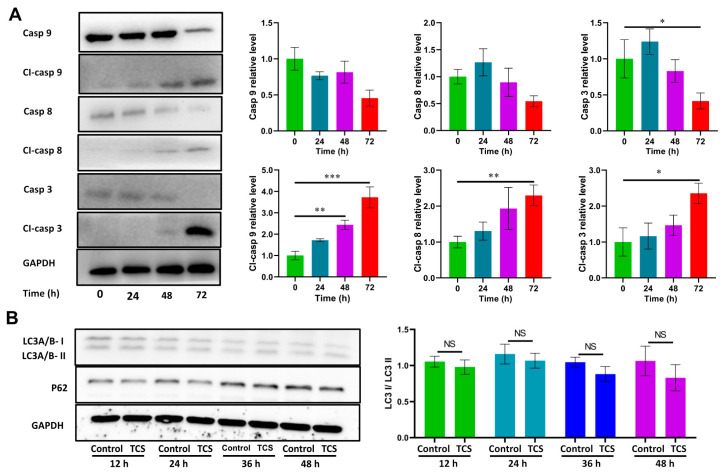
Effects of TCS on apoptosis and autophagy of HCC cells. (**A**) H22 HCC cells were treated with 25 μg/mL TCS for 24 h, 48 h and 72 h. Western blot assayed the levels of key apoptosis proteins Caspase 9, Cleaved-caspase 9, Caspase 8, Cleaved-caspase 8, Caspase 3 and Cleaved-caspase 3; (**B**) H22 HCC cells were treated with 25 μg/mL TCS for 12 h, 24 h, 36 h and 48 h. Western blot assayed the levels of key autophagy proteins P62 and LC3A/B. NS means no significant difference, *, *p* < 0.05; **, *p* < 0.01; ***, *p* < 0.001.

**Figure 3 ijms-24-01416-f003:**
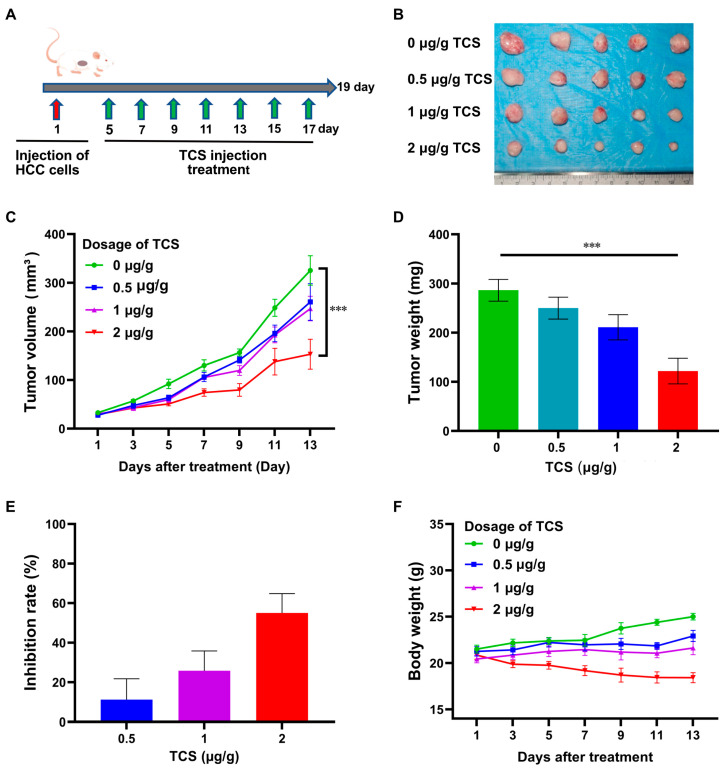
Effects of TCS on apoptosis and autophagy of HCC cells. (**A**) Timeline schedule of mice injected with HCC cells and treated with TCS; (**B**) On day 19, mice were euthanized and tumor tissues were obtained as shown in the figure; (**C**–**E**) Quantitation of data showed the volume of tumors, weight and the inhibition rate of TCS on the weight of tumor tissue; (**F**) Quantitation of data showed the weight changes in mice treated with varying doses of TCS. ***, *p* < 0.001.

**Figure 4 ijms-24-01416-f004:**
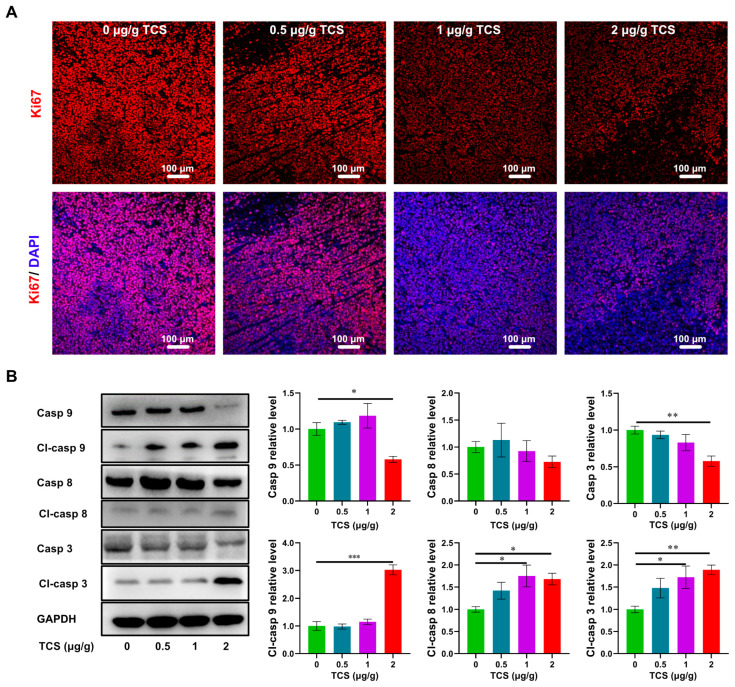
Effects of TCS on proliferation and apoptosis of HCC in vivo. (**A**) Immunofluorescence method detected nuclear proliferation factor Ki67 (Ki67: red, DAPI: blue). Bar = 100 μm; (**B**) Western blot method detected the protein levels of Caspase 9, Cleaved-caspase 9, Caspase 8, Cleaved-caspase 8, Caspase 3 and Cleaved-caspase 3 in HCC tissues. *, *p* < 0.05; **, *p* < 0.01; ***, *p* < 0.001.

**Figure 5 ijms-24-01416-f005:**
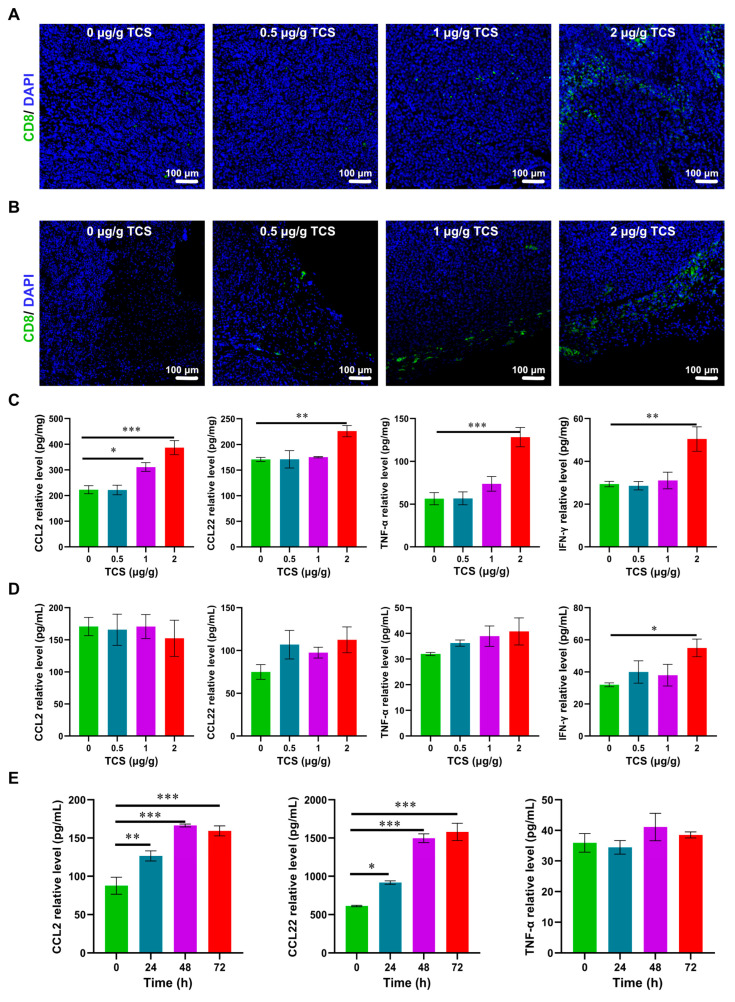
Effects of TCS on chemotactic enrichment. (**A**) Immunofluorescence method detected CD8-positive cells in the center of HCC tissues. Bar = 100 μm; (**B**) Immunofluorescence method detected CD8-positive cells at the edge of HCC tissues. Bar = 100 μm; (**C**) ELISA method detected the protein levels of CCL2, CCL22, TNF-α and IFN-γ in HCC tissues treated with different doses of TCS (0, 0.5, 1 and 2 μg/g); (**D**) ELISA method detected the protein levels of CCL2, CCL22, TNF-α and IFN-γ of mouse serum; (**E**) ELISA method detected the protein levels of CCL2, CCL22 and TNF-α in H22 HCC cells treated with 25 μg/g TCS after 24 h, 48 h and 72 h, GAPDH as the internal reference gene. *, *p* < 0.05; **, *p* < 0.01; ***, *p* < 0.001.

**Figure 6 ijms-24-01416-f006:**
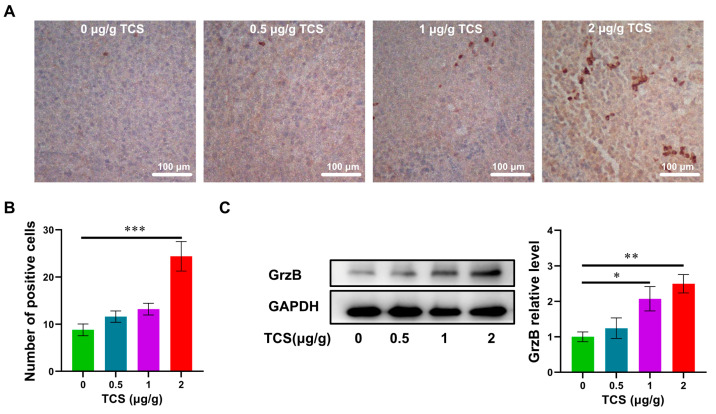
Effects of TCS on GrzB expression in vivo. (**A**,**B**) Immunochemical method detected GrzB in hepatocellular carcinoma tissues. Bar = 100 μm; (**C**) Western blot method detected the level of GrzB in HCC tissues treated with different doses of TCS. *, *p* < 0.05; **, *p* < 0.01; ***, *p* < 0.001.

**Figure 7 ijms-24-01416-f007:**
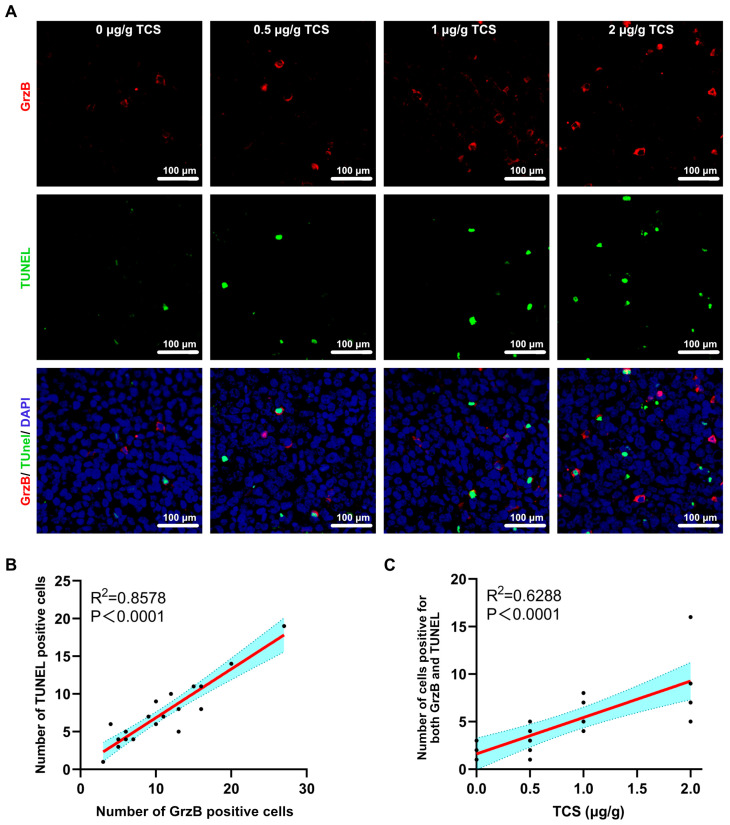
TCS promotes GrzB-induced apoptosis in hepatocellular carcinoma. (**A**) Immunochemical method detected TUNEL and GrzB in HCC tissues after TCS treatment. Bar = 100 μm; (**B**) Correlation analysis of GrzB positivity with the number of positive signals for TUNEL; and (**C**) Correlation analysis of the number of simultaneous positive signals for GrzB and TUNEL with the dose of TCS.

**Figure 8 ijms-24-01416-f008:**
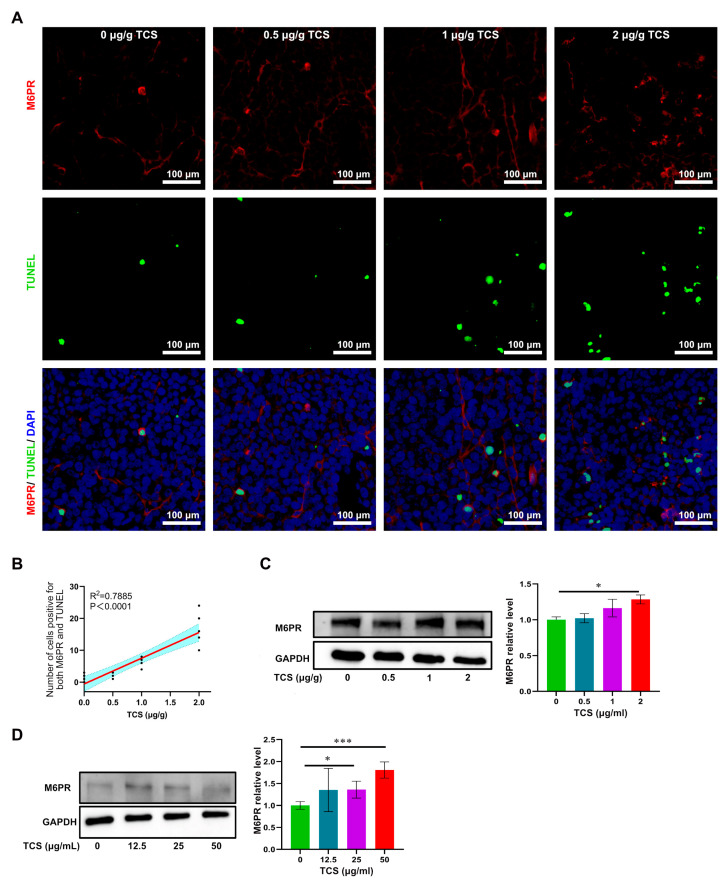
TCS promotes M6PR expression in vivo and in vitro. (**A**) Immunochemical method detected GrzB and M6PR in HCC tissues upon TCS treatment. Bar = 100 μm; (**B**) Correlation analysis of the number of simultaneous positive signals of GrzB and M6PR with the dose of TCS; (**C**) Western blot method detected the levels of M6PR in HCC tissues treated with different doses of TCS (0, 0.5, 1 and 2 μg/g); (**D**) Western blot method detected the levels of M6PR in H22 HCC cells treated with different doses of TCS (0, 12.5, 25 and 50 μg/mL). *, *p* < 0.05; ***, *p* < 0.001.

**Figure 9 ijms-24-01416-f009:**
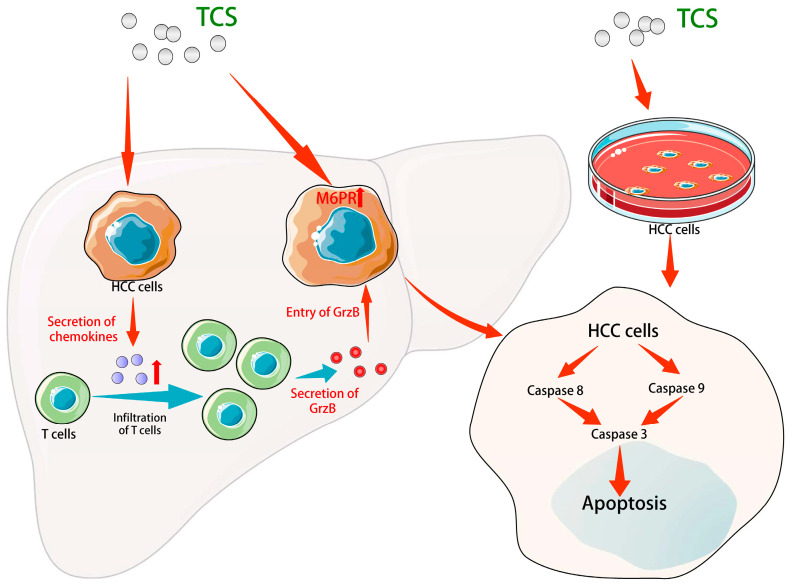
Diagram showing the molecular mechanism of TCS on anti-tumor activity. TCS stimulated the expression of chemokines CCL2, CCL17 and CCL22, which may encourage the enrichment of CD8^+^ T cells within HCC tissue. GrzB secreted by the T cells were transported into the HCC cells by cellular M6PR and facilitated cell apoptosis. TCS-induced caspases mediated apoptosis both in vivo and in vitro.

## Data Availability

Not applicable.

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
