# Peer review of "Trichosanthin Promotes Anti-Tumor Immunity through Mediating Chemokines and Granzyme B Secretion in Hepatocellular Carcinoma"

_ijms, 2023, doi:10.3390/ijms24021416_

Round 1

Reviewer 1 Report

The manuscript by Wang and colleagues deals with induction of T-cell infiltration and immune response against hepatocellular carcinoma by the plant protein trichosanthin. The antitumor properties of this protein and its mechanisms of action are well studied. The novelty of this manuscript is the use of a hepatocellular carcinoma model. The authors found that trichosanthin induces recruitment of CD8+T cells in the tumor tissue and increased levels of the chemokines CCL2, CCL17 and CCL22. In addition, they confirmed the well-known mechanisms of action of trichosanthin: apoptosis and involvement of granzyme B. The manuscript is well illustrated.

There are some confusions in the manuscript that need to be resolved before publication.

-        Authors talk about T-cell chemokines, but in Figure 5 they show that H22 (hepatocarcinoma cells) have increased expression levels of the investigated chemokines after treatment with trichosanthin. So, which cells express CCL2, CCL17 and CCL22?

-        May be the authors well know that the increased levels of mRNA does not mean that these chemokines will be secreted. In this case, to prove the mechanism of action in the in vivo model, it is better to measure the serum levels of secreted cytokines.

Other minor points:

Please, provide the clones, manufacturer and the conjugation of the antibodies used in the study.

How was extracted and prepared the trichosanthin or what is the source (manufacturer) of the trichosanthin? This information is missing in the section Materials and Methods.

Line 91: “… HCC cells ….. were extracted for genes…..” This sentence has to be rewritten.

Line 298: Please, provide the source of the cell line H22.

Line 299: Are you sure that the units for the penicillin are ug/mL?

Line 310: “…enzyme marker…” ?!? This sentence has to be rewritten.

Line 396: The TSC is not PHYCOprotein drug.

Author Response

Dear Reviewer,

Thank you very much for reviewing our manuscript and all the valuable comments and suggestions. We have carried out the experiments that your suggested and revised the manuscript accordingly. We have incorporated the feedbacks into our revised manuscript highlighted in yellow and responded to each suggestion point by point as below. We hope that you find our responses satisfactory.

Comment 1: Authors talk about T-cell chemokines, but in Figure 5 they show that H22 (hepatocarcinoma cells) have increased expression levels of the investigated chemokines after treatment with trichosanthin. So, which cells express CCL2, CCL17 and CCL22?

Response 1: We thank the reviewer for raising the valuable question. Studies have reported that tumor cells, macrophages and NK cells can all produce the investigated chemokines [1-3]. In this study, we showed that tumor cells treated with TCS produced increased levels of the investigated chemokines in vitro

  1. Henry, C. J., D. A. Ornelles, L. M. Mitchell, K. L. Brzoza-Lewis and E. M. Hiltbold. "Il-12 produced by dendritic cells augments CD8+ T cell activation through the production of the chemokines CCL1 and CCL17." J Immunol 181 (2008): 8576-84. 10.4049/jimmunol.181.12.8576.
  2. Rapp, M., S. Grassmann, M. Chaloupka, P. Layritz, S. Kruger, S. Ormanns, F. Rataj, K. P. Janssen, S. Endres, D. Anz, et al."C-C chemokine receptor type-4 transduction of T cells enhances interaction with dendritic cells, tumor infiltration and therapeutic efficacy of adoptive T cell transfer." Oncoimmunology 5 (2016): e1105428. 10.1080/2162402x.2015.1105428.
  3. Fialová, A., S. Partlová, L. Sojka, H. Hromádková, T. Brtnický, J. Fučíková, P. Kocián, L. Rob, J. Bartůňková and R. Spíšek. "Dynamics of T-cell infiltration during the course of ovarian cancer: The gradual shift from a Th17 effector cell response to a predominant infiltration by regulatory T-cells." Int J Cancer 132 (2013): 1070-9. 10.1002/ijc.27759.

Comment 2:  May be the authors well know that the increased levels of mRNA does not mean that these chemokines will be secreted. In this case, to prove the mechanism of action in the in vivo model, it is better to measure the serum levels of secreted cytokines.

Response 2: Thank you for this valuable feedback. Yes, the increased levels of mRNA does not mean that these chemokines will be secreted. Therefore, following your great suggestion, we further examined the protein levels of four cytokines, CCL2, CCL22, TNF-α and IFN-γ which were significantly changed at the level of mRNA, in cell culture supernatants, xenograft mouse serum and tumor tissues by ELISA. The expression levels of chemokines CCL2 and CCL22, as well as TNF-α and IFN-γ in tumor tissue were significantly increased upon TCS treatment. Serum levels of CCL22, TNF-α and IFN-γ was elevated after TCS treatment, although only IFN-γ statistically significantly increased. In addition, the expression levels of chemokines CCL2 and CCL22 in HCC cell culture fluid were also significantly increased upon TCS treatment. The ELISA results have been added to Figure 5C-5E, and the corresponding result descriptions have been modified.

Other minor points:

 Comment 1: Please, provide the clones, manufacturer and the conjugation of the antibodies used in the study.

Response 1: Thank you very much for your valuable comment. We have fully elaborated the clones, manufacturer and the conjugation of the antibodies used in the study within Table S1. 

Comment 2: How was extracted and prepared the trichosanthin or what is the source (manufacturer) of the trichosanthin? This information is missing in the section Materials and Methods.

Response 2: Thank you for your precious advice. We have added the information of “Recombinant TCS protein extraction and purification” to the module of “Materials and Methods” of the manuscript (lines 341-352).

 Comment 3: Line 91: “… HCC cells ….. were extracted for genes…..” This sentence has to be rewritten.

Response 3: We appreciate your valuable suggestion. We have rewritten this sentence and revised the corresponding part (lines 94-96) of the manuscript.

 Comment 4: Line 298: Please, provide the source of the cell line H22.

Response 4: Thank you very much for your precious comments. We have elaborated relevant information about the source of the cell line H22 and added it to line 353 of the manuscript.

Comment 5: Line 299: Are you sure that the units for the penicillin are ug/mL?

Response 5: Thank you for your kind recommendation. We have corrected the units for the penicillin from ug/mL to U/mL in the manuscript (line 355).

 Comment 6: Line 310: “…enzyme marker…” ?!? This sentence has to be rewritten.

Response 6: We appreciate your valuable suggestion. We have rewritten this sentence and revised it accordingly in the manuscript (lines 364-366).

 Comment 7: Line 396: The TCS is not PHYCOprotein drug.

Response 7: We are so grateful for your kind comments. According to your suggestion, we have amended “The TCS is PHYCOprotein drug” to “The TCS is a natural drug” in the manuscript (line 458).

Reviewer 2 Report

1- the source of TSC is not listed

2- age, weight, and sex of animals should be included

3- the rationale for using selected doses!

4-some missing signs for statistical differences of some figures

5- ANOVA should be used instead of t-test for multiple comparisons followed by an appropriate posthoc test

Author Response

Dear Reviewer,

We appreciate your thorough reading of this manuscript and for the constructive suggestions, which greatly help to improve the quality of this manuscript. We have incorporated the feedbacks into our revised manuscript highlighted in yellow and responded to each suggestion point by point as below. We hope that you find our responses satisfactory.

Comment 1: the source of TCS is not listed

Response 1: Thank you for your valuable comments. We have added the information of “Recombinant TCS protein extraction and purification” to the module of “Materials and Methods” of the manuscript (lines 341-352).

Comment 2: age, weight, and sex of animals should be included

Response 2: We are so grateful for your kind suggestion. “Male BALB/c mouse, 5 weeks old, 20±2 g (Guangdong Medical Laboratory Animal Center, Guangzhou)” has been mentioned in the manuscript (lines 393-394).

Comment 3: the rationale for using selected doses!

Response 3: The drug dose selection in animal experiments was based on the preliminary experiment of this study, and multiple doses was tested. The dose of 2 μg/g showed excellent tumor inhibition effect. But when the dose reaches 4 μg/g or higher, it showed certain toxic effect on mouse liver. Therefore, only 2 μg/g TCS were used in the experiment. The cell drug dose was selected based on the CCK-8 assay that detected an IC50 of about 25 μg/mL.

Comment 4: some missing signs for statistical differences of some figures

Response 4: Thank you very much for your precious comments. Statistical charts with differences of figures have been added (Figure 1F).

Comment 5: ANOVA should be used instead of t-test for multiple comparisons followed by an appropriate posthoc test.

Response 5: We appreciate your precious advice. The relevant statistical graph has been modified accordingly. In addition, we also modified the “Statistical analysis” in the module of “Materials and Methods” in the manuscript (lines 447-452).

Reviewer 3 Report

In the manuscript entitled “Trichosanthin promotes anti-tumor immunity through mediating chemokines and granzyme B secretion in hepatocellular carcinoma” the authors evaluated in vitro and in vivo activities of Trichosanthin.

 Even though the subject is relevant and the results are new, this reviewer believes that the manuscript possesses some quality issues, which raised the following concerns:

 Minor concerns:

1-   The scientific name (Juniperus communis) It's not in italic. And it should be.

2- Sometimes microgram per mL is written with L in lower case and sometimes in upper case. Please uniform it.

Major concerns:

1-   According to the authors in the materials and methods section, each experiment was repeated three times. But the raw data files only show one experiment. Some with the total membrane, and some are cropped membranes.

2-      In the line 86, the authors stated that “Z-VAD-FMK significantly inhibited the cell death and PARP cleavage”, but it is not clear how the authors evaluated the significance, since the “cleaved PARP” band is weak. And considering that, according to the authors, the experiment was repeated three times, this reviewer believes that this was the best image of the three. This issue is reinforced by the fact the in the next figures (2 and 4) the authors present densitometry for the W.B. bands. Please, clarify.

3-      The importance of chemokines (CCL2, CCL17 and CCL22) was poorly discussed.

4-      In line 227 It is written “A recent study showed that tonics containing the Chinese herbal medicine smallpox significantly…”. What this mean?

Based on these concerns this reviewer DOES NOT RECOMMEND publication in the International Journal of Molecular Sciences, in the present form.

Author Response

Dear Reviewer,

Thank you very much for your precious time in reviewing our manuscript (ijms-2102185), entitled “Trichosanthin promotes anti-tumor immunity through mediating chemokines and granzyme B secretion in hepatocellular carcinoma” and for your valuable comments. We have addressed your comments, and the amendments are highlighted in yellow in the revised manuscript. We hope that the revision is acceptable, and we look forward to hearing from you soon.

Minor concerns:

Comment 1: The scientific name (Juniperus communis) It's not in italic. And it should be.

Response 1: Thank you for your precious advice. We have revised the Juniperus communis to Trichosanthes in the manuscript (line 50).

Comment 2: Sometimes microgram per mL is written with L in lower case and sometimes in upper case. Please uniform it.

Response 2: Thank you very much for your valuable propose. We have modified the manuscript according to your suggestion.

Major concerns:

Comment 1: According to the authors in the materials and methods section, each experiment was repeated three times. But the raw data files only show one experiment. Some with the total membrane, and some are cropped membranes.

Response 1: Thank you for your valuable advice. We have uploaded the original data from three independent experiments. Due to the different experimental habits of the experimental operators, some experimental results showed the total membranes while others cropped membranes. And, cropped membranes in the same experiment all come from the same membrane.

Comment 2: In the line 86, the authors stated that “Z-VAD-FMK significantly inhibited the cell death and PARP cleavage”, but it is not clear how the authors evaluated the significance, since the “cleaved PARP” band is weak. And considering that, according to the authors, the experiment was repeated three times, this reviewer believes that this was the best image of the three. This issue is reinforced by the fact the in the next figures (2 and 4) the authors present densitometry for the W.B. bands. Please, clarify.

Response 2: Thank you for your precious suggestion. We are very sorry for the misunderstanding because we did not demonstrate the best results. Therefore, we have replaced Figure 1F with a better one.

Comment 3: The importance of chemokines (CCL2, CCL17 and CCL22) was poorly discussed.

Response 3: Thank you for the suggestion. We have added the information required as explained as follows: “Tumor-infiltrating lymphocytes are an important prognostic factor for cancer progression and a key player in cancer immunotherapy. CCL2, CCL17 and CCL22 are cytokines with a role in T-cell recruitment. Despite the role of CCL2 in recruiting both cytotoxic T cells (CTL) and monocytes to tumor sites, studies have shown that enhancement of the CCL2/CCR2 axis or inhibition of CCL2 nitration in antitumor therapy significantly promotes T cell infiltration and exerts antitumor effects. binding of CCL22 with CCR4 enhances T cell dendritic cell binding and increases CTL activation, while enhancing tumor cell responses to IFN-γ.”

Comment 4: In line 227 It is written “A recent study showed that tonics containing the Chinese herbal medicine smallpox significantly…”. What this mean?

Response 4: We are so grateful for your question and deeply sorry for the confusion. We have modified it as follows: “A recent study showed that tonics containing the Chinese herb Trichosanthes significantly elevated serum levels of cytokines such as IFN-γ, IL-6, and TNF-α, which would be beneficial in enhancing the immune effect of the lymphocyte system.”

Round 2

Reviewer 1 Report

Please, replace "T cell chemokines" in the text with "chemokines"

Reviewer 3 Report

The authors properly adressed  my concerns. Thus, I RECOMMEND this paper for publication in  IJMS.